# Performance of a New Grouting Material under the Coupling Effects of Freeze–Thaw and Sulfate Erosion

**DOI:** 10.3390/ma16155448

**Published:** 2023-08-03

**Authors:** Qinyong Ma, Biao Li

**Affiliations:** 1School of Civil Engineering and Architecture, Anhui University of Science and Technology, Huainan 232001, China; qymaah@126.com; 2Engineering Research Center of Underground Mine Construction, Ministry of Education, Anhui University of Science and Technology, Huainan 232001, China

**Keywords:** grouting material, sulfate attack, freezing and thawing cycle, microstructure

## Abstract

In order to study the performance of a new cement-based grouting material under the coupling of freeze–thaw cycle and sulfate erosion, tests related to the performance of the new grouting material were designed and carried out to analyze the damage mechanism of the material under the coupling of freezing and thawing and Na_2_SO_4_ solution by testing the mass change, relative dynamic elastic modulus, compressive strength loss and mineralogical and microstructural properties of the new grouting material. The test results show that with the increase in the number of freeze–thaw cycles, the mass loss and compressive strength loss of the specimens in 15% Na_2_SO_4_ solution gradually increased, and the relative dynamic elastic modulus showed a decreasing trend. When the freeze–thaw cycle number was 30, the mass loss rate, compressive strength loss rate and relative dynamic elastic modulus of the specimens in Na_2_SO_4_ solution were 4.17%, 24.59% and 84.3%, respectively, which showed better erosion and frost durability. Mineralogical and microstructural analysis showed that SO_4_^2−^ in solution led to the decomposition of the C-S-H gel and the formation of CaSO_4_•2H_2_O inside the specimen, and the internal deterioration was exacerbated by the widening of the crack width being aggravated, suggesting that the rate of material deterioration under the coupling of the two factors increased.

## 1. Introduction

The 21st century is the era of great development of underground space, and the scale of underground engineering construction is expanding. Due to the vast territory of China, a large number of underground engineering structures have been in a harsh environment for a long time. Especially in the northwest of China, there are a large number of salt marsh soil and salt lakes. Soil and groundwater contain a large amount of SO_4_^2−^ ions. One of the most serious factors affecting underground engineering’s long-term durability is sulfate erosion [1]. Additionally, the temperature difference between day and night in this area is large, and freeze–thaw damage affects a project’s durability. Cement-based grouting materials have been widely used in underground engineering to control groundwater damage and reinforce soft soil layers [2]. Therefore, research on the durability of cement-based grouting materials is extremely important for improving the safety and durability of underground engineering structures. 

Zhu et al. [3] studied the durability and water dispersion resistance of high-performance synchronous grouting materials and concluded that a reduction in the water–binder ratio or a reduction in bentonite content and the incorporation of a certain proportion of functional composite admixtures can significantly improve the durability and water dispersion resistance of grouting materials. Jiao et al. [4] analyzed the corrosion resistance index of grouting materials with different water–cement ratios, and the results showed that the strength corrosion coefficient of the grouting material with a water–cement ratio of 0.65 was greater than 0.80 after immersion in a saturated magnesium sulfate solution for 360 days, showing good corrosion resistance. Liu et al. [5] investigated the effect of gypsum on the performance of alkali slag–fly ash slurry under composite excitation and concluded that under the action of freeze–thaw cycles, the value of the loss of strength of the nodule body was positively correlated with the gypsum dosage, and the loss of strength was the most significant in the first five freeze–thaw cycles. Zhao et al. [6] investigated the water stability and frost resistance of a new grouting material for foam lightweight soil, and the results showed that the strength of the specimen decreased with the increase in the number of freeze–thaw cycles, and the larger the water–cement ratio, the weaker the frost resistance of the specimen. Wang et al. [7] investigated the sulfate resistance of grouting slurries mixed with lead–zinc tailings. The results showed that in the early stage of erosion, the increase in sulfate concentration was beneficial for the increase in strength, but with the prolongation of soaking time, the higher the concentration of the sulfate solution, the more serious the decrease in the strength of the slurry caking body. Wang et al. [8] investigated the effect of metakaolin (MK) on the resistance of grouting materials to sulfate and sulfuric acid erosion. The results showed that the grouting material with an MK doping of 20% had high durability against sulfate and sulfuric acid erosion. Ding et al. [9] studied the sulfate corrosion resistance of cement paste mixed with granulated blast furnace slag (GBFS). The results showed that the addition of GBFS could effectively resist the decalcification and dealumination of C-A-S-H gel by sulfate and reduce the expansion damage of ettringite secondary formation to the stone body. Liu et al. [10] studied the linear expansion, mass change, mechanical properties and microstructure of alkali-activated slag grouting material (AASGM) after soaking in different concentrations of sodium sulfate solution for 90 days. The results show that the main gel product of the AASGM sample is C-A-S-H gel, the structure expansion is slight, the mass increase is small, the compressive strength is basically unchanged and the sulfate corrosion resistance is good.

The above research results show that research scholars have obtained many research results on the resistance of grouting materials to sulfate attack and freeze–thaw damage, but there are few studies on the performance of cement-based grouting materials under the combined action of sulfate attack and freeze–thaw cycles. Therefore, the freeze–thaw test of a new cement-based grouting material mixed with ultra-fine ground granulated blast furnace slag and silica fume was carried out, and the properties of the specimen under different freeze–thaw cycles in an aqueous solution and sulfate solution were studied, in order to provide a reference for practical engineering.

## 2. Test

### 2.1. Raw Material

Ordinary Portland cement (OPC) with a strength grade of PO42.5 produced by Bagongshan Cement Plant in Huainan City was selected. Ultra-fine ground granulated blast furnace slag (UFS) was produced by Wuhan Huashen Intelligent Technology Co., Ltd. (Wuhan, China) Silica fume (SF) was produced by Henan Yuanheng Environmental Protection Engineering Co., Ltd. (Henan, China). PCE is a polycarboxylate superplasticizer produced by Shanghai Kaiyuan Chemical Technology Co., Ltd. (Shanghai, China), and the mixed water was ordinary tap water. The particle size and particle size distribution of raw materials were measured using a BT-2003 laser particle size distribution instrument. Figure 1 shows the particle size distribution of raw materials. It can be seen from Figure 1 that the D_50_ of OPC, SF and UFS is 17.57 μm, 9.13 μm and 3.99 μm, respectively, with good particle gradation. Table 1 shows their chemical composition. 

### 2.2. Test Scheme

The effects of incorporating different mass fractions of UFS, SF and PCE at different water–cement ratios on the properties of the grouting materials were investigated using orthogonal tests and polar analysis of variance, and the optimal mix ratios for the grouting materials were obtained through comprehensive equilibrium analyses, i.e., water–cement ratio of 0.70, 20% (mass fraction) UFS, 12% (mass fraction) SF and 0.16% (mass fraction) PCE [11]. The performance test results of the optimally proportioned grout are shown in Table 2.

The specimen was made using a cement glue sand mixer; firstly, the weighed cement, UFS and SF were poured into the mixer and subjected to dry mixing for 1–2 min until the mixture was homogeneous, and then the accurately weighed PCE and water were mixed evenly, poured into the mixer and mixed for 2–3 min. The prepared slurry was poured into a three-link plastic mold with a side length of 70.7 mm, without vibration, and then moved into the standard curing room for 24 h after demolding, and the demolded specimens were placed in the curing room for the freezing and thawing test after water curing to 90 days age. After demolding, the specimen was placed in the maintenance room for 90 days for the freeze–thaw cycle test. The relative humidity in the curing room was not less than 95%, the temperature was easily controlled at 20 ± 2 °C, the water temperature was easily controlled at 20 ± 1 °C and the depth of water on the upper surface of the specimen during the curing period was not less than 5 mm.

The freeze–thaw cycle test was divided into an aqueous solution freeze–thaw and salt solution freeze–thaw, and the salt solution was 15% Na_2_SO_4_ solution by mass fraction. The freezing and thawing cycle test method was selected with reference to the “ordinary concrete long-term performance and durability performance test method” (GB/T 50082-2009) [12] in the anti-freezing test of the slow-freezing method, the test apparatus was an STDW-40D-type high- and low-temperature test chamber. According to the specification requirements, the samples were tested every 25 cycles; however, due to the small size of the sample, the frost resistance of the paste was slightly poor, so the number of freeze–thaw cycles was set to 0, 5, 15, 30 and 50. In each freeze–thaw cycle, the temperature was controlled at −20 °C and maintained for 5 h during freezing in the high- and low-temperature chamber, and then the sample was removed and thawed at room temperature (20 °C) for 7 h, i.e., 12 h per cycle. After reaching the specified number of times, the mass, compressive strength and ultrasonic velocity of the sample were tested, and the mass loss rate, strength loss rate and relative dynamic elastic modulus of the sample were calculated to judge the frost resistance of the new grouting material.

### 2.3. Test Method

#### 2.3.1. Basic Performance Test Method for Slurry

The basic working performance test of grouting slurry refers to the “Technical specifications for cement grouting construction of hydraulic buildings” (DL/5148-2021), and the specific test steps are as follows.

The bleeding rate test was carried out with a measuring cylinder, the prepared slurry was poured into a 100 mL cylinder, the initial height value H_1_ of the slurry was recorded and sealed, and after 24 h of resting, the height value H_2_ of the slurry secretion in the cylinder was recorded, and the ratio of H_2_ to H_1_ was the slurry water secretion rate. The slurry viscosity was determined using the standard funnel method. The 1006 slurry viscometer was selected. A finger was used to block the lower end of the funnel. The prepared slurry was poured into the standard funnel through the filter until it was flush with the funnel mouth. The 500 mL measuring cup mouth was facing upward, and the finger was removed to make the slurry in the funnel flow into the 500 mL measuring cup. The time taken for the slurry to flow fully into the measuring cup was the viscosity of the slurry, and the unit was s. The fluidity test was carried out using a truncated cone die with an upper diameter of 36 mm and a lower diameter of 60 mm and a glass plate of 450 mm × 450 mm × 5 mm.

#### 2.3.2. Mass Loss Rate

In this test, an electronic balance with an accuracy of 0.01 g and a range of 3000 g was used to determine the mass of the specimens. There were 3 parallel specimens in each group, and the average value was taken as the mass measurement in g. The formula used to determine the rate of mass loss is as follows:(1)$$w_{i}=\frac{M_{0}-M_{i}}{M_{0}}$$

The formula comprises three variables: $w_{i}$, which denotes the mass loss rate of the specimen following *i* freeze–thaw cycles; $M_{0}$, which represents the mass of the specimen prior to undergoing freeze–thaw cycles; and $M_{i}$, which signifies the mass of the specimen after *i* freeze–thaw cycles.

#### 2.3.3. Relative Dynamic Elastic Modulus

An NM-4B non-metallic ultrasonic detection analyzer was used for the test. The test procedure was as follows: Firstly, wipe the two probes and the surface of the specimen clean, and then evenly apply petroleum jelly on the surface of the specimen; select ultrasonic detection; set the measurement distance to 70.7 mm, the sampling period to 0.1 us and the emission voltage to 250 V; and use the two probes to clamp the two ends of the specimen, so that the center of the specimen is at the same horizontal line as far as possible. When the sine wave appears on the screen, click the sampling button to record the propagation time T/us. Five data are measured for each group of specimens and the average value is obtained. The relative dynamic elastic modulus $E_{rd}$ [13] is calculated as follows:(2)$$E_{rd}=\frac{E_{dt}}{E_{d0}}=\frac{V_{t}^{2}}{V_{0}^{2}}=\frac{T_{0}^{2}}{T_{t}^{2}}$$

In the formula, *V*, *T* and *t* represent ultrasonic sound velocity (km/s), ultrasonic sound time (us) and freeze–thaw times (times), respectively.

#### 2.3.4. Compressive Strength Loss Rate

The RMT rock mechanics testing machine was selected to determine the compressive strength of the specimens. The force (large) stroke control was used, and the loading rate was set at 0.1 km/s. The compressive test results of the specimens were averaged from three parallel specimens to an accuracy of 0.1 MPa. The following is the calculation formula for compressive strength loss rate $\;f_{c}$:(3)$$f_{c}=\frac{f_{c0}-f_{cn}}{f_{c0}}\times 100\%\;$$

In the formula, $f_{c0}$ and $f_{cn}$ represent the compressive strength before the freeze–thaw cycle and the compressive strength after *n* freeze–thaw cycles, respectively.

#### 2.3.5. Mineralogical and Microstructural Analysis

Specimens were taken from the central part of the compressive specimens, and the hydration was terminated by soaking in anhydrous ethanol. Before the test, the samples were taken out and put in an oven for drying at 55 °C for 6 h. 

(1)XRD analysis

The dried specimen was finely ground in an agate mortar. Finally, the powder was passed through a 200-mesh sieve, and the powder under the sieve was taken for testing. A Smartlab SE X-ray diffractometer was selected as the test instrument. The Cu target was used for the test. The diffraction angle (2θ) was 5–80°, and the step size was 5°/min.

(2)SEM analysis

A FlexSEM1000 scanning electron microscope manufactured by Hitachi of Tokyo, Japan was selected as the test instrument. The main equipment was filament, the test condition was vacuum mode, the test voltage was 15~20 kv and the working distance was 5 nm~10 nm. The specimen size of 5 mm × 5 mm × 5 mm after drying was selected for testing, and the specimen was sprayed with gold before testing.

(3)TG analysis

The thermogravimetric test instrument was a Mettler Toledo TGA2 from Greifensee, Switzerland. After drying, the specimen was processed into powder, placed in an alumina crucible, and tested in a nitrogen atmosphere, with the temperature ranging from room temperature to 1000 °C and the heating rate being 10 °C/min.

## 3. Results

### 3.1. Appearance of Damage in Specimens

Table 3 shows the appearance of damage in the specimens under different freeze–thaw cycles in aqueous solution and 15% Na_2_SO_4_ solution. 

Before the number of freeze–thaw cycles reaches 30, whether the specimen is in aqueous solution or Na_2_SO_4_ solution, the appearance of the specimen remains basically intact, and the surface of the specimen is complete and undamaged with a small amount of pitting and holes. When the number of freeze–thaw cycles reaches 30, the specimen surface pits increase, a local surface peeling phenomenon occurs, the specimen in Na_2_SO_4_ solution has more slag on the surface, and cracks appear on the edges and corners. When the number of freeze–thaw cycles reached 50 times, the corners of the specimens in Na_2_SO_4_ solution were corroded and rounded and a through crack appeared on the surface, the surface of the specimens in aqueous solution peeled off quite seriously, and the exposure of the internal aggregates was more obvious, but the original corners were still kept and the surface did not have cracks.

### 3.2. Mass Loss Rate

Figure 2 shows the mass change law of the specimens after freeze–thaw damage in aqueous solution and 15% Na_2_SO_4_ solution.

As can be seen from Figure 2, the mass loss rate of the specimen with the increase in the number of freeze–thaw cycles shows an increasing trend. When the number of freeze–thaw cycles is 15, the mass loss rate of 1.30% for the specimen in the Na_2_SO_4_ solution is reduced by 24% compared to that for the specimen in the aqueous solution (1.71%), which is due to the presence of sulfate, to a certain extent, reducing the freezing point of the water, increasing the compressibility of ice, and playing a role in inhibiting the freezing and thawing cycles [13]. When the number of freeze–thaw cycles reached 30, the mass loss rate of the specimens in aqueous and Na_2_SO_4_ solutions reached 3.94% and 4.17%, respectively. As the number of freeze–thaw cycles increased, the mass loss rate of the specimens after freeze–thaw cycles in Na_2_SO_4_ solution was gradually higher than the value of freeze–thaw damage in aqueous solution, which was due to the fact that the high concentration of sulfate would greatly shorten the time of inhibition of the damage inhibition effect of freezing and thawing on the specimens [14], and with the freeze–thaw cycles and sulfate attack, the surface of the specimens appeared to have cracks, and the corners were corroded with rounded corners, and there was more slagging, so the quality of the specimens continued to be reduced, and the rate of mass loss was increased. After 50 freeze–thaw cycles, the specimens reached a mass loss rate of 12.6% for freeze–thaw cycles in aqueous solution and 12.63% for the combined effect of freeze–thaw and Na_2_SO_4_ solution erosion, both of which exceeded 5%, and the freeze–thaw cycle test was completed.

### 3.3. Relative Dynamic Modulus of Elasticity

Figure 3 shows the E_rd_ change of specimens subjected to freeze–thaw damage in two different solutions.

As can be seen in Figure 3, the E_rd_ of the specimens after freeze–thaw cycles in both aqueous and Na_2_SO_4_ solutions showed a decreasing trend, and the E_rd_ in Na_2_SO_4_ solution was higher than the value of freeze–thaw cycling in an aqueous solution; the reason for this is that the SO_4_^2−^ ions will enter into the interior of the specimen to react with the alkaline substances such as Ca(OH)_2_ produced by the hydration of the cement, and the erosion product generated fills up the internal pore space and makes the specimen more compact [15]. When the number of freeze–thaw cycles reached 30, the E_rd_ of the specimens in aqueous and Na_2_SO_4_ solutions was 79.92% and 84.3%, respectively, and the decrease reached 20.09% and 15.7%, respectively. When the number of freeze–thaw cycles reached 50, the test was terminated because the mass loss rate of the specimen exceeded 5%; at this time, the E_rd_ of the specimens was 75.38% and 79.38%, respectively, which were both above 75% of the initial value, and it can be seen that the impact of freeze–thaw cycles on the quality of cementitious grouting materials is greater than the impact on the E_rd_; that is, the erosion damage to the surface of the specimen is more serious than the damage to the internal structure.

### 3.4. Compressive Strength Loss Rate

Figure 4 shows the change rule of compressive strength loss in the specimens after different freeze–thaw cycles in the two solutions. As can be seen from Figure 4, the compressive strength of the specimens showed a decreasing trend with the increase in the number of freeze–thaw cycles; before 30 freeze–thaw cycles, the compressive strength of the specimen in Na_2_SO_4_ solution is greater than that in aqueous solution, after 30 freeze–thaw cycles, the compressive strength curve of the specimen decreases significantly, and when 50 freeze–thaw cycles are completed, the compressive strength of the specimen in aqueous solution and in Na_2_SO_4_ solution decreases to 11.45 MPa and 9.15 MPa, respectively.

It can be seen from Figure 5 that the compressive strength loss rate of the specimen increases with the increase in the number of freeze–thaw cycles. When the number of freeze–thaw cycles reaches 30, the compressive strength loss rate of the specimen in aqueous solution is higher than that in Na_2_SO_4_ solution. With the increasing number of freeze–thaw cycles, the specimen compressive strength loss rate of the law of change is the opposite. The reason for this is that in the pre-freezing and thawing period, the presence of sulfate largely reduces the freezing point of water and increases the compressibility of ice, which plays an inhibitory role in the internal damage of the specimen, and the sulfate ions will react with the hydration products of the cement, and the eroded substances generated will fill the internal pores and make the structure more compact, so the loss rate of compressive strength of the specimen in the pre-freezing and thawing period in the Na_2_SO_4_ solution is lower than that of the specimen in the aqueous solution. However, with the increase in the number of freeze–thaw cycles, the expansion stresses caused by the sulfate erosion products together with the freeze–thaw frost heave effect aggravate the internal deterioration of the material, leading to an accelerated reduction in compressive strength [16]. After 50 freeze–thaw cycles, the compressive strength loss rate of the specimen in aqueous solution and Na_2_SO_4_ solution reached 68.93% and 75.17%, respectively, indicating that the combined effect of sulfate and freeze–thaw in the later stage of the test deepened the internal damage degree of the grouting material.

### 3.5. Uniaxial Compressive Stress–Strain Curve

Figure 6 shows the uniaxial compressive stress–strain curves of the specimens under the action of freezing and thawing in the two solutions. It can be seen that the change trends of the stress–strain curves of the samples after different freeze–thaw times in aqueous solution and Na_2_SO_4_ solution are basically the same. As the number of freeze–thaw cycles increases, the slope of the rising section of the stress–strain curve gradually decreases, the peak stress point of the curve decreases, and the curve gradually becomes flat. The strain corresponding to the peak stress reached by the specimen in the aqueous solution is larger than that in the Na_2_SO_4_ solution because the sulfate reacts with the hydration products of the cement, and the erosion products generated fill the pores and make the internal structure denser, whereas the specimen in the aqueous solution has a relatively large number of pores and therefore results in a larger deformation at a smaller stress.

### 3.6. Mineralogical and Microstructural Analysis

#### 3.6.1. XRD Analysis

Figure 7 shows the XRD patterns of the specimens under freeze–thaw cycles in the two solutions. As can be seen in Figure 7, the main hydration products of the specimens after freeze–thaw cycling in aqueous solution are Ca(OH)_2_, AFt (ettringite) crystals, C-S-H gel, CaCO_3_, etc. Ding et al. investigated the effect of freeze–thaw cycling on the microscopic properties of new cementitious materials and came to the same conclusion [17]. The Ca(OH)_2_ diffraction peaks of the samples in the aqueous solution increased with the number of freeze–thaw cycles compared to the pre-freeze–thaw period. This is mainly due to the relative increase in Ca(OH)_2_ with the freeze–thaw cycles as hydration continues to occur within the sample. The main hydration products and aqueous solution types of the specimens after freeze–thaw cycles in Na_2_SO_4_ solution were essentially the same, with enhanced AFt diffraction peaks, significantly reduced C-S-H diffraction peaks and the generation of gypsum diffraction peaks compared to the freeze–thaw of aqueous solutions. The reason for this is that SO_4_^2−^ can enter the interior through the pores on the surface of the specimen and form AFt crystals with aluminum-phase hydration products in materials such as cement, and also react with Ca(OH)_2_ to form CaSO_4_•2H_2_O [18], but it can lead to the decomposition of C-S-H and cause the leaching of calcium ions [19].

#### 3.6.2. TGA Analysis

The results of the specimens’ TG-DTG analysis following freeze–thaw cycles in aqueous solution and Na_2_SO_4_ solution are shown in Figure 8. As can be seen from the DTG curves in Figure 8, there is a large peak in the DTG curves at around 100 °C due to the loss of water from AFt, C-S-H gel and free water [20,21], and there are two peaks at around 400 °C and 700 °C, which are mainly due to the dehydration of Ca(OH)_2_ [22] and CaCO_3_ [23]. The TG curves show an increasing trend in the percentage mass loss of the specimens with increasing temperature. Compared with the pre-freeze–thaw period, the mass loss of the specimens in aqueous solution increased with the number of freeze–thaw cycles, and the mass loss in Na_2_SO_4_ solution for 30 freeze–thaw cycles approximately overlapped with the pre-freeze–thaw TG curves, and the maximum mass loss was reached for 50 freeze–thaw cycles. This is due to the fact that SO_4_^2−^ ions enter the interior through the pores on the surface of the specimen and react inversely with the cement hydration product Ca(OH)_2_ to produce a small amount of AFt crystals, but a large amount of AFt crystals and CaSO_4_•2H_2_O are produced at a later stage. After a high temperature of 800 °C, the mass loss of the specimens after 50 freeze–thaw cycles in aqueous solution and Na_2_SO_4_ solution was 23.28% and 23.73%, respectively.

#### 3.6.3. SEM Analysis

The samples frozen and thawed 30 times and 50 times in two solutions were selected for an SEM test, and the test photos are shown in Figure 9. As can be seen from Figure 9e, the specimen has hexagonal lamellar Ca(OH)_2_ crystals in the interior before freezing and thawing and shows a distribution of a large number of C-S-H gels, which have a relatively dense structure. It can be seen from Figure 9a that Ca(OH)_2_ crystals and C-S-H gel were distributed inside the structure when the sample was subjected to 30 freeze–thaw cycles in aqueous solution, but some tiny holes appeared. As the number of freeze–thaw cycles continued to increase, a large number of C-S-H gels were still distributed inside the specimens, and no other types of substances were produced, but an increase in the number of interlocking cracks and holes produced inside the structure can be observed in Figure 9b, which was attributed to the fact that with the increase in the number of freeze–thaw cycles, the larger freezing and expansion forces caused cracks to appear inside the specimens. As can be seen from Figure 9c, compared to the specimens in the aqueous solution, when the specimens were frozen and thawed in Na_2_SO_4_ solution 30 times, the internal products produced pin-columnar AFt, filling the internal pores and tiny cracks, with a more dense structure, which was macroscopically manifested as an increase in strength. When the specimens were subjected to 50 freeze–thaw cycles in Na_2_SO_4_ solution, a large amount of AFt and a small amount of C-S-H gel were distributed in the internal cracks of the structure, and the width of the internal cracks widened. This is because SO_4_^2−^ ions will react with the hydration products in the pores to form CaSO_4_•2H_2_O and AFt with expansive properties, which enlarges the pores or cracks [24] and therefore is macroscopically manifested as a reduction in strength, which further explains the accelerated deterioration rate of the new cement-based grouting material under the combined effect of sulfate and freeze–thaw cycles.

## 4. Conclusions

(1)When the specimens were subjected to 50 freeze–thaw cycles in an aqueous solution and sodium sulfate solution, the mass loss rate was more than 5%, but the relative dynamic elastic modulus was more than 75% of the initial value, indicating that the freeze–thaw cycles had a greater impact on the quality of cement-based grouting materials than on the relative dynamic elastic modulus; that is, the surface erosion damage of the material samples was more serious than the damage to the internal structure.(2)Before the number of freeze–thaw cycles reaches 30, the compressive strength loss rate of the specimens in the two solutions is less than 30%, showing good freeze resistance. After 50 freeze–thaw cycles, the compressive strength loss rate of the samples in aqueous solution and Na_2_SO_4_ solution reached 68.93% and 75.17%, respectively, indicating that the combined action of sulfate and freeze–thaw cycles deepened the internal damage degree of the grouting materials in the late test period.(3)Mineralogical and microstructural analysis shows that SO_4_^2−^ in the solution leads to the decomposition of C-S-H gel and the formation of CaSO_4_•2H_2_O in the sample, and the widening of crack width aggravates the internal deterioration of the structure, indicating that the deterioration rate of grouting material under freeze–thaw and sulfate erosion increases.

## Figures and Tables

**Figure 1 materials-16-05448-f001:**
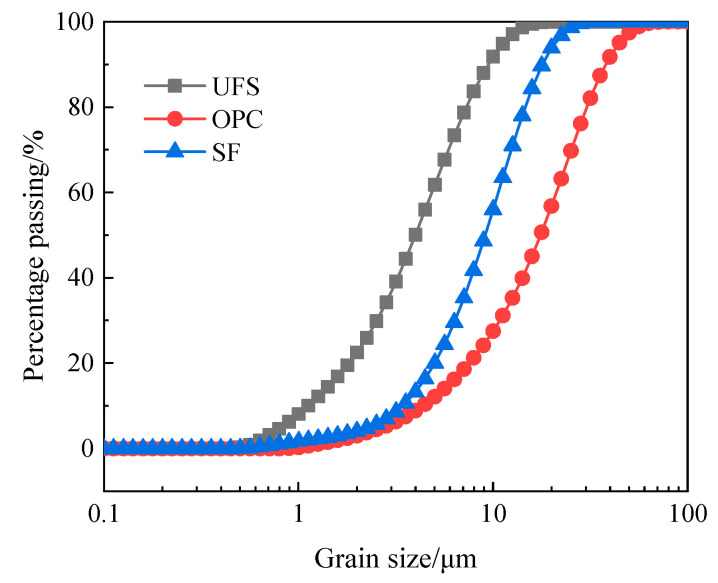
Particle size distribution of raw materials.

**Figure 2 materials-16-05448-f002:**
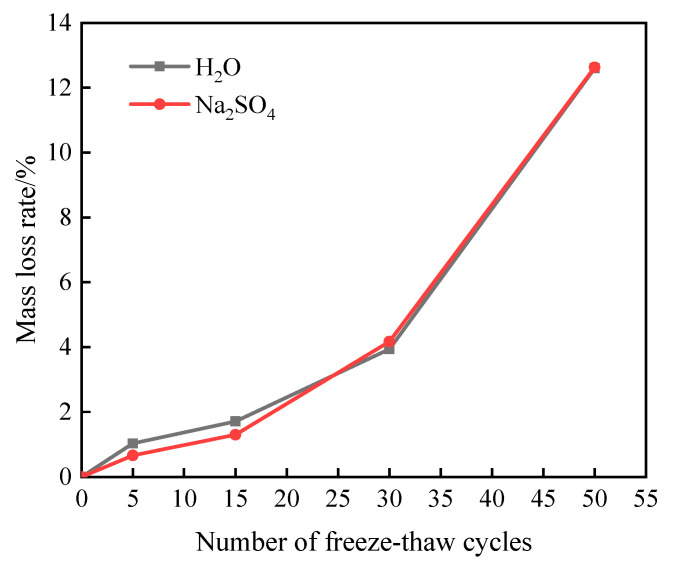
Quality change of freeze–thaw damage specimen.

**Figure 3 materials-16-05448-f003:**
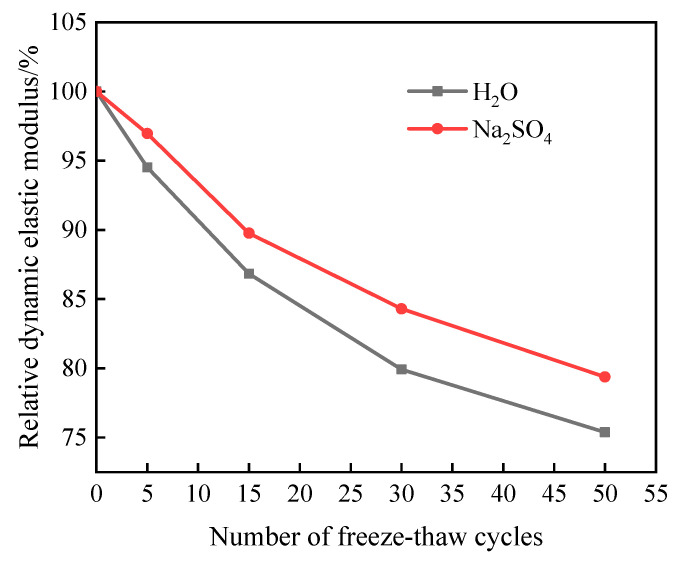
The relative dynamic elastic modulus change of freeze–thaw damage specimen.

**Figure 4 materials-16-05448-f004:**
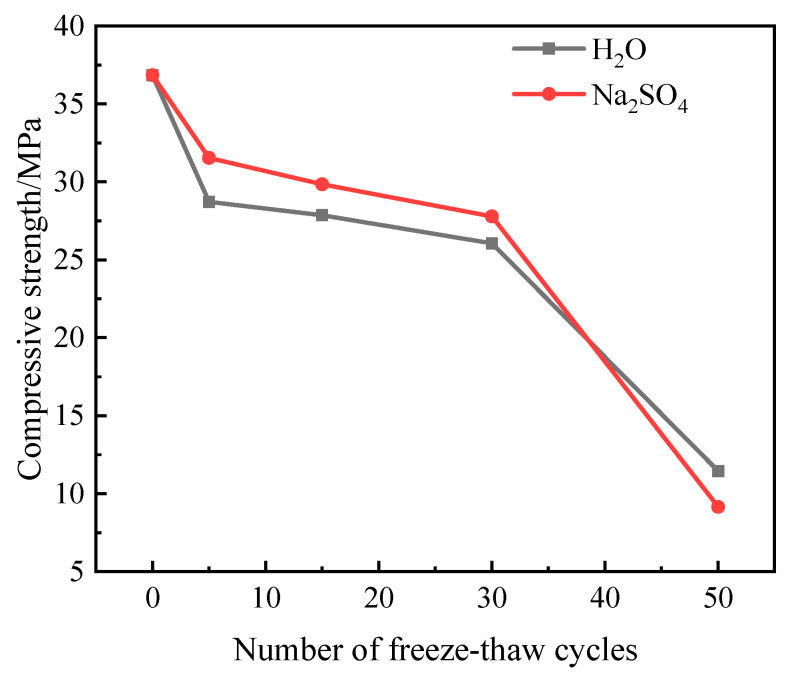
Change in compressive strength of freeze–thaw damage specimen.

**Figure 5 materials-16-05448-f005:**
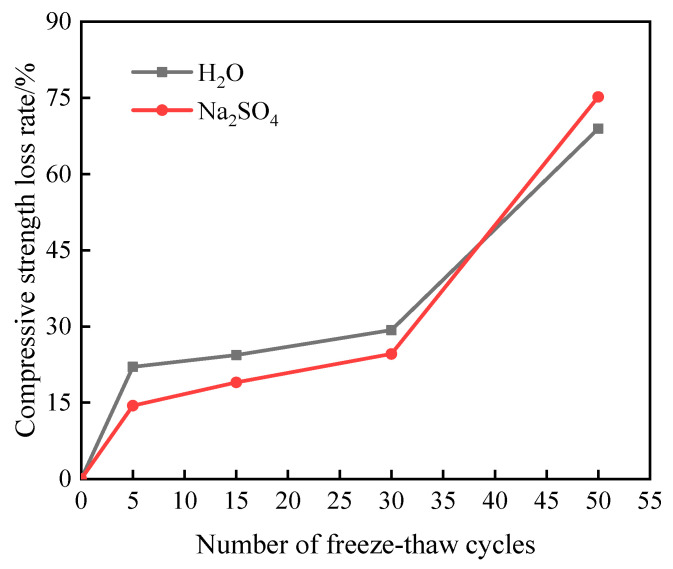
The change in compressive strength loss rate of freeze–thaw damage specimen.

**Figure 6 materials-16-05448-f006:**
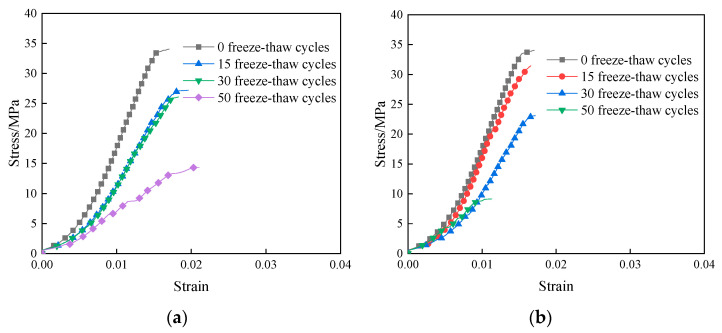
Uniaxial stress–strain curve of specimen under freeze–thaw cycle: (**a**) H_2_O; (**b**) Na_2_SO_4_.

**Figure 7 materials-16-05448-f007:**
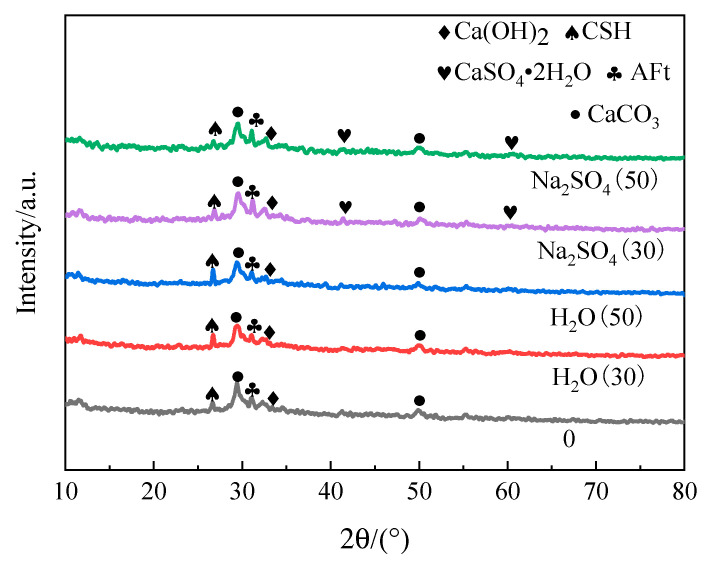
XRD patterns of freeze–thaw damage specimens.

**Figure 8 materials-16-05448-f008:**
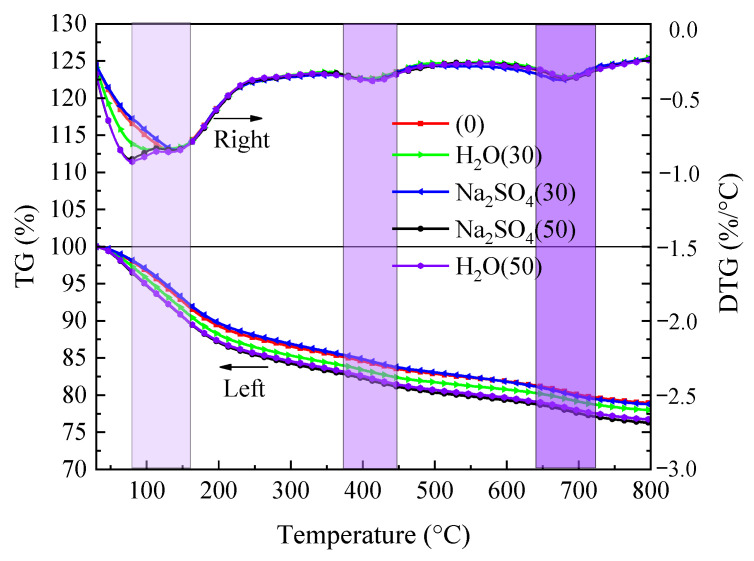
Analysis of TG and DTG of samples under freeze–thaw cycle.

**Figure 9 materials-16-05448-f009:**
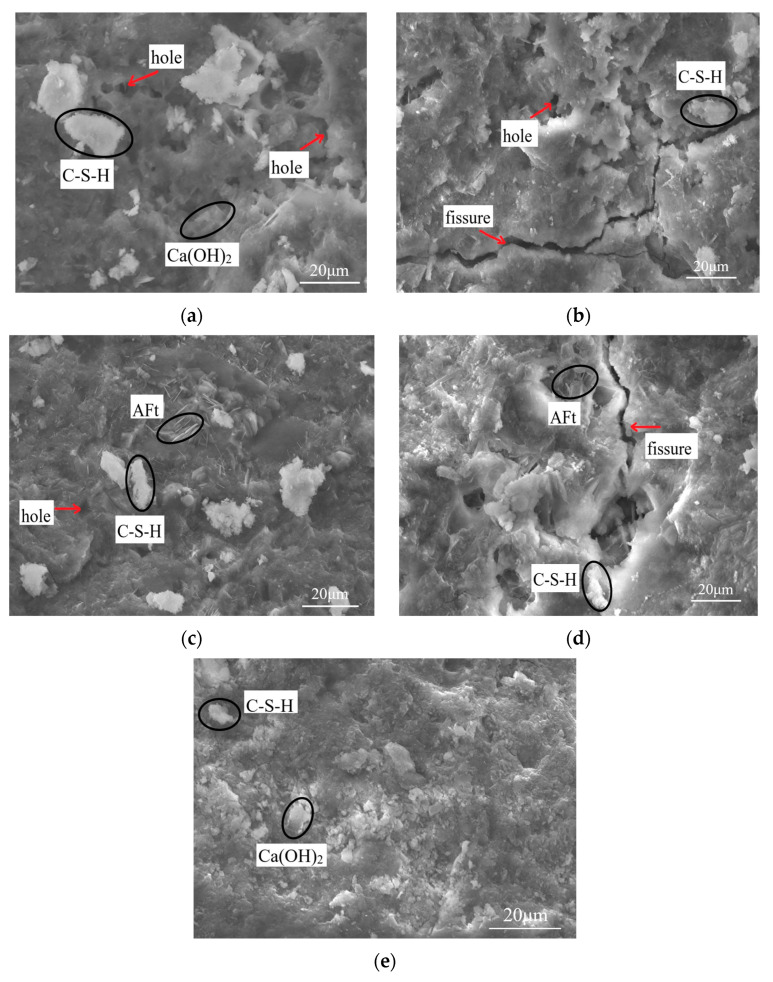
SEM photos of freeze–thaw damage specimens: (**a**) H_2_O (30 freeze–thaw cycles); (**b**) H_2_O (50 freeze–thaw cycles); (**c**) Na_2_SO_4_ (30 freeze–thaw cycles); (**d**) Na_2_SO_4_ (50 freeze–thaw cycles); (**e**) before the freeze–thaw cycle.

**Table 1 materials-16-05448-t001:** The chemical composition of raw materials.

Raw Material	Mass Fraction (%)									
	CaO	SiO_2_	Al_2_O_3_	Fe_2_O_3_	MgO	Na_2_O	K_2_O	SO_3_	TiO_2_	Loss
OPC	66.3	19.60	6.50	3.50	0.70	0.60	0.30	2.50	—	—
UFS	39.25	33.40	15.15	0.31	7.67	0.38	0.39	2.38	0.62	0.11
SF	0.10	96.46	0.31	0.07	0.11	0.97	—	—	—	2.60

**Table 2 materials-16-05448-t002:** Performance test results of optimally proportioned grout.

Performance Index	Bleeding Rate (%)	Fluidity (mm)	Viscosity (s)	Compressive Strength (MPa)			Flexural Strength (MPa)		
				3 Days	7 Days	28 Days	3 Days	7 Days	28 Days
Optimized slurry	1.6	307	33.46	9.7	15.9	27.3	4.5	5.2	7.5

**Table 3 materials-16-05448-t003:** Appearance of damage in specimens under different freeze–thaw cycles.

	Number	0	10	15	30	50
Solution						
Na_2_SO_4_		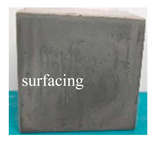	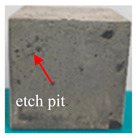	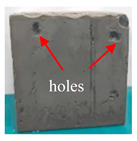	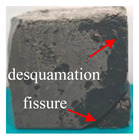	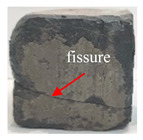
H_2_O			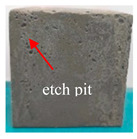	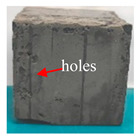	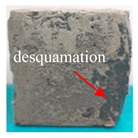	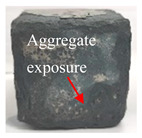

## Data Availability

The data that support the findings of this study are available from the corresponding author upon reasonable request.

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
