# Peer review of "Performance of a New Grouting Material under the Coupling Effects of Freeze–Thaw and Sulfate Erosion"

_materials, 2023, doi:10.3390/ma16155448_

Round 1

Reviewer 1 Report

The manuscript »Performance of a new grouting material under the coupling effects of freeze-thaw and sulfate erosion« is interesting but some parts should be improved before the publication. The novelty of the study should be better annotated. The  following corrections are also suggested:

-Line 24: Introduction should be improved.

- Line 41: Subscript in chemical formula.

-Line 64: Is »P•O42.5« correct?

-  66-68: Superscript in units for specific surface area and density. Also the description of methods which were used for measuring specific surface area and density of materials are missing, as well as the used instruments and details of potential sample  preparation – please add. Another important parameter of the raw materials is the granulometry. Please add also particle size distribution data (PSD), the method/instrument/settings/sample preparation/measurement medium also needs to be described (laser granulometry).

-Line 70: Please describe the method which was used for determing chemical composition of the samples. The used instrument also needs to be stated, including details of sample preparation.

-Line 72: Please specify units in Table 1 – did you mean weight %?

- Line 73-75: The reference (9) is not available, neither in Chinese nor in English. The raw material proportions should be stated in a more clear way.

- Line 77: The methods from Table 2 should be described in text (standards/ instruments /main settings). Some units are incorrect, please check.

- Line 80: Such photos of the raw materials are not needed (give no important information). Instead of them PSD diagrams (cumulative curve, distribution density) should be presented.

- Line 85: Please specify standard curing roomparameters (T,R.H.). Dry curing? Were  moulds uncovered/unsealed?

- Line 115: Sample photos on Figure 2 are too small – it is hard to see the damage. In figure captions the used solutions should be mentioned.

- Figures 3-6: In figure captions the used solutions should be mentioned (you are comparing samples, immersed in two different solutions).

- Line 127-146: Subscripts in chemical formulas.

- Line 100: All methods, mentioned in sections  3 and 4, should be listed and described in section 2. Materials and Methods, including instruments and main settings and sample preparation.

- Line 213: Figure 7: In figure captions the numbers 0, 15, 30 and 50 (different curves) should be explained.

- Line 214: Microstructural analysis  should be a subsection of the Results. RD should be described in section 2. Materials and Methods, including instruments and main settings and sample preparation. For XRD, the cards, which were used for phase identification, should be also  given (ICDD database codes...).

- Line 219-222: Subscripts in chemical formulas.

- Line 227: Intensity axis has arbitrary units, XRD-patterns are one above other (offset) and  the units 1-8 on y-axis are not correct and not needed (please delete them). The way of writing phases in the legend should be the same as in the text of the MS.  You can use abbreviations on figure and explain them in figure captions. The meaning of XRD pattern labels should be also explained in figure captions.

- Line 228: See comment from Line 100.

-Line 243: The legend should be explained in figure captions.

- Line 247: Subscript.

- Line 245-263: »Microscopic holes« is not an appropriate term. However, porosity is important parameter which could improve interpretation of the results, but it should determined by an appropriate method. Based on SEM micrographs from Figure 10 it is not possible to identify phases. The scale bars are also not clear and some descriptions are in Chinese alplhabet.

Line 267: Conclusions should be improved.

Author Response

Point 1: The introduction of line 24 should be improved and the subscript of the chemical formula in line 41 is incorrect, and the "P•O42.5" in line 64 is incorrect.

Response 1: Thank you very much for the valuable comments put forward by the reviewers, which have been improved in the revised draft, and the modified part has been marked in red.

Point 2: The specific surface area and density units in lines 66-68 are marked with errors, the method used to measure the specific surface area and density of the material is not yet stated, and a description of the method used to measure the specific surface area and density of the material is also missing, the instrument used and details of the sample preparation. Another important parameter of the raw material is the particle size, please also add the particle size distribution data (PSD), the method/instrument/setup/sample preparation/measurement medium also needs to be described (laser particle size meter).

Response 2: Thanks very much for the reviewer’s valuable advice. Since the specific surface area and density of the material are provided by the manufacturer, the measurement method is not introduced. The specific surface area and density of the raw material have been deleted. Laser particle size measurement of raw materials has been carried out, as detailed in the revised draft, and the modified part has been marked in red.

Point 3: Please indicate the unit in table 1 on line 72.

Response 3: Thanks for the valuable comments of the reviewers, which have been indicated in the revised draft, see the revised draft for details, and the revised part has been marked in red.

Point 4: The reference (9) in lines 73-75 is neither in Chinese nor in English. The proportion of raw materials should be more clearly stated.

Response 4: It has been modified in the revised draft, and the modified part has been marked in red, see the revised draft for details.

Point 5: The methods in Table 2 should be described in text (standard/instrument/main setup). Some units are incorrect, please check.

Response 5: Thank you very much for the valuable comments put forward by the reviewers, which have been improved in the revised draft, and the modified part has been marked in red.

The test methods are as follows:

The bleeding rate was tested with a measuring cylinder, the prepared slurry was poured into a 100 mL measuring cylinder, the initial height of the slurry H1 was recorded, and it was sealed; After standing for 24h, the height value H2 of serous bleeding in the measuring cylinder was recorded, and the ratio of H2 to H1 was the bleeding rate of serous fluid. Standard funnel method was used to measure slurry viscosity, 1006 mud viscometer was selected, the flow outlet of the funnel was blocked with fingers, the prepared slurry was poured into the standard funnel until it was level with the funnel mouth, the mouth of the 500 mL measuring cup was pointed upwards, and the finger was removed to make the slurry in the funnel flow into the 500 mL measuring cup. The time it takes for the slurry to fill the 500 mL measuring cup is the viscosity of the slurry, expressed in s. (This method uses the funnel type to measure the relative viscosity of the slurry, so the unit is s, if the rotary viscometer is used to measure the absolute viscosity of the slurry, the unit is Pa·s.) Slurry flow was tested according to DL/T 5148-2012. A truncated cone die with an upper diameter of 36 mm and a lower diameter of 60 mm and a glass plate with a diameter of 450 mm×450 mm×5 mm were used to test the flow.

Point 6: Delete Figure 1, the photo sample in Figure 2 is too small to see the damage, Please specify standard curing roomparameters.

Response 6: Figure 1 The raw material sample was changed to a particle size distribution map, and Figure 2 specimen damage diagram has been modified, please refer to the revised draft for details. The relative humidity in the curing room is not less than 95%, the temperature is easy to control at 20±2℃, the water temperature is easy to control at 20±1℃, and the water depth on the upper surface of the sample should not be less than 5mm during the curing period.

Point 7:  Figures 3-6: In figure captions the used solutions should be mentioned (you are comparing samples, immersed in two different solutions). All methods, mentioned in sections  3 and 4, should be listed and described in section 2. Materials and Methods, including instruments and main settings and sample preparation.

Response 7: Thanks for the valuable comments of the reviewers, it has been revised in the corresponding place, see the revised draft for details.

Point 8: In figure captions the numbers 0, 15, 30 and 50 (different curves) should be explained. Microstructural analysis  should be a subsection of the Results. XRD should be described in section 2. Materials and Methods, including instruments and main settings and sample preparation.

Response 8: The numbers 0, 15, 30 and 50 in the figure represent different freeze-thaw cycles respectively, for example, 15 represents 15 freeze-thaw cycles. Microscopic XRD test methods and instruments have been added, see the revised draft for details, and the modified part has been marked in red.

Point 9:  Intensity axis has arbitrary units, XRD-patterns are one above other (offset) and  the units 1-8 on y-axis are not correct and not needed (please delete them). The way of writing phases in the legend should be the same as in the text of the MS.  You can use abbreviations on figure and explain them in figure captions. The meaning of XRD pattern labels should be also explained in figure captions.

Response 9: Thanks to the valuable comments of the reviewers, the numbers on the y axis in the XRD pattern have been deleted, and the content introduction has been modified. For details, see the revised draft, and the modified part is marked in red.

Point 10: “Microscopic holes” is not an appropriate term. However, porosity is important parameter which could improve interpretation of the results, but it should determined by an appropriate method. Based on SEM micrographs from Figure 10 it is not possible to identify phases. The scale bars are also not clear and some descriptions are in Chinese alplhabet.

Response 10: Thanks to the valuable comments of the reviewers, the inappropriate terminology has been changed, the picture has been changed to 2000 times, the ruler has been marked clearly in the picture, see the revised draft for details.

Point 11: Conclusions should be improved.

Response 11: Thanks to the valuable comments of the reviewers, the conclusion has been revised as follows:

(1) When the samples were freeze-thaw cycles reached 50 times in aqueous solution and sodium sulfate solution, the mass loss rate was more than 5%, but the relative dynamic elastic modulus was more than 75% of the initial value, indicating that the freeze-thaw cycle had a greater impact on the quality of cemency-based grouting materials than on the relative dynamic elastic modulus, that is, the surface erosion damage of the material samples was more serious than the damage to the internal structure.

(2) Before the freeze-thaw cycle reaches 30 times, the compressive strength loss rate of the samples in the two solutions is less than 30%, showing good freeze-resistance. After 50 freeze-thaw cycles, the compressive strength loss rate of the samples in aqueous solution and Na2SO4 solution reached 68.93% and 75.17%, respectively, indicating that the combined action of sulfate and freeze-thaw deepened the internal damage degree of the grouting materials in the late test period.

(3) Microscopic analysis shows that SO42- in the solution leads to the decomposition of C-S-H gel and the formation of Gyp in the sample, and the widening of crack width aggravates the internal deterioration of the structure, indicating that the deterioration rate of grouting material under freeze-thaw and sulfate erosion increases.

Reviewer 2 Report

The introduction is well written and presents the readers with the toppic of the article.

Materials and methods has to be significantly improved as it lacks description of methodology for most of the tests conducted (XRD, TGA, SEM, strength testing). In addition to these please provide the specific surface area and density also for OPC as it is the only raw material without these characteristics.

In general the discussion in parts 3 and 4 has to be improved as there are insufficient references to literature, comparisons with behaviour of similar materials in similar conditions etc. Also while reading the article, the further form the introduction the less clear are the statements made and the sentences comprehensible.

The typography has to be improved, check for correct display of chemical formulas, units (also please use degree sign and to the word degree). Use abbreviations where possible (e.g. Erd instead of 'relative elastic dynamic modulus').

L 128 - The sentence staring there does not describe the results presented in the figure.

L 162 - What the 'current' mean? After 50 cycles? unclear.

L 168-170 The first two sentences say exactly the same only in different words, so please use just one.

L 188 - sentence starting there please rephrase, it's hard to keep in touch with authors' intended idea.

L 208-212 - The combination of the sentences implies that more dense samples have more ?holes?. Please rephrase and use adequate terminology

L 268 The sentence ends with semicolon and no conclusion.

Figure 10 replace the chinese descriptions with english or remove.

Conclusions only highlight the results but do not provide with some overall conclusion like is the tested material better/worse than currently used? Generally try to answer questions/goals set in the introduction.

Please check and rephrase the text for 'tortured sentences', comprehensibility of the statements etc.

Author Response

Point 1: Materials and methods has to be significantly improved as it lacks description of methodology for most of the tests conducted (XRD, TGA, SEM, strength testing). In addition to these please provide the specific surface area and density also for OPC as it is the only raw material without these characteristics.

Response 1: Thank you very much for the valuable comments of the reviewers. The method introduction and test steps have been added in the corresponding position of the revised draft. However, due to the limited test equipment of the school, the specific surface area and density of the material were not tested, and some data were provided by the merchants who provided the product, so the density and specific surface area of the raw material were uniformly deleted. Please refer to the revised draft for details. The modified part has been marked in red.

Point 2: In general the discussion in parts 3 and 4 has to be improved as there are insufficient references to literature, comparisons with behaviour of similar materials in similar conditions etc. Also while reading the article, the further form the introduction the less clear are the statements made and the sentences comprehensible.

Response 2: Thanks very much for the valuable opinions of the reviewers, the part of the test results has been revised, please see the revised draft for details, and the revised part has been marked in red. However, due to the limited literature on the study of sulfate and freeze-thaw cycles on cementing grouting materials, there is no large number of references, and the literatures cited are all related materials.

Point 3: The typography has to be improved, check for correct display of chemical formulas, units (also please use degree sign and to the word degree). Use abbreviations where possible (e.g. Erd instead of 'relative elastic dynamic modulus').

Response 3: Thank you very much for the valuable comments of the reviewers. The font, chemical formula and units of the paper have been carefully modified. The abbreviation for the description of the relative dynamic elastic modulus is also used, as shown in the revised draft.

Point 4: The experimental results in the figure are not described in line 128.

Response 4: Thank you very much for the reviewer's valuable comments. “The quality loss rate of the sample in Na2SO4 solution is lower than that in aqueous solution after 30 freeze-thaw cycles” has been revised to “When the number of freeze-thaw cycles is 15 times, the quality loss rate of the sample in Na2SO4 solution is 1.30, which is 24% lower than that in aqueous solution of 1.71”. Please refer to the revised draft for details. The changes have been highlighted in red.

Point 5: Line 162: What the 'current' mean? After 50 cycles? unclear.

Response 5: This place has been modified to read: “When the freeze-thaw cycles reached 50 times, the test was terminated because the mass loss rates of the samples immersed in aqueous solution and Na2SO4 solution exceeded 5%. At this time, the Erd of the samples were 75.38% and 79.38%, respectively, both of which were above 75% of the initial values.”

Point 6: Lines 168-170: The first two sentences say exactly the same only in different words, so please use just one.

Response 6: Thank you very much for the valuable comments made by the reviewers, the “Figure 5 demonstrates how the sample 's compressive strength decreases as the number of freeze-thaw cycles rises. ” has been deleted, as detailed in the revised version, the modified part has been marked red.

Point 7: Lines 208-212: The combination of the sentences implies that more dense samples have more ?holes?. Please rephrase and use adequate terminology.

Response 7: This place has been carefully modified to read: “The strain corresponding to the peak stress of the sample in the aqueous solution is larger than that in the Na2SO4 solution. The reason is that the sulfate will react with the cement hydration products, and the generated erosion products fill the pores and the structure is more dense. In the aqueous solution, the sample has relatively more pores, so under the smaller stress, it will lead to larger deformation.”

Point 8: Conclusions only highlight the results but do not provide with some overall conclusion like is the tested material better/worse than currently used? Generally try to answer questions/goals set in the introduction.

Response 8: The conclusion has been modified as follows:

(1) When the samples were freeze-thaw cycles reached 50 times in aqueous solution and sodium sulfate solution, the mass loss rate was more than 5%, but the relative dynamic elastic modulus was more than 75% of the initial value, indicating that the freeze-thaw cycle had a greater impact on the quality of cemency-based grouting materials than on the relative dynamic elastic modulus, that is, the surface erosion damage of the material samples was more serious than the damage to the internal structure.

(2) Before the freeze-thaw cycle reaches 30 times, the compressive strength loss rate of the samples in the two solutions is less than 30%, showing good freeze-resistance. After 50 freeze-thaw cycles, the compressive strength loss rate of the samples in aqueous solution and Na2SO4 solution reached 68.93% and 75.17%, respectively, indicating that the combined action of sulfate and freeze-thaw deepened the internal damage degree of the grouting materials in the late test period.

(3) Microscopic analysis shows that SO42- in the solution leads to the decomposition of C-S-H gel and the formation of Gyp in the sample, and the widening of crack width aggravates the internal deterioration of the structure, indicating that the deterioration rate of grouting material under freeze-thaw and sulfate erosion increases.

Round 2

Reviewer 1 Report

Line 85: It would be more transparent if the title of the chapter would indicated which material the testing methods refer to.

Line 81: “%” is mass %? Please specify (Point 3 in previous MS version).

Line 91: Please unify the capitalization in Table 2 and check the units for fluidity and viscosity.

Point 5: Partly corrected. Some additional comments regarding Point 5:

- The chapter title 2.3.5 “Microscopic test” is an unusual expression, as well as XRD and TGA are not performed by microscope.  It would be maybe better to say “Mineralogical and microstructural analysis” or similar. Similar comment for 3.6 Microstructure analysis (line 262), where it would be also more correct to say “Mineralogical and microstructural analysis”.

- For XRD please add main instrument setup and sample preparation (were the samples dry, granulometry of analysed samples) and  list the powder diffraction file (PDF) codes used for phase determination.

- For SEM please add the instrument information (producer, model) and main settings (filament, vacuum mode (low or high vacuum) an accelerating voltage, working distance etc.) and sample preparation (any coating?).

Point 7: Figure captions of Figures 2-5 (in the revised version) were not corrected.

Line 266: Please add some discussion (comparison with other authors) for the sentence “Following freezing and thawing in aqueous solution, Figure 7 shows that the sample’s primary hydration products are Ca(OH)2, AFt (ettringite) crystal, C-S-H gel, and CaCO3.”

Line 280: Figure 7 captions should be improved. In legend either chemical formula either phase name should be used (now is mixed), please unify. XRD peaks of gypsum are not clear. The diagram might be cut at 55 or 60°2 theta in order to make peaks more clear.

Point 10: Based on SEM micrographs from Figure 9 it is still hard to identify phases. Some labels are incomprehensible (different alphabet) and should be corrected.

Line 332: At point (3) at the beginning of the sentence instead of »Microscopic analysis.« the term “Mineralogical and microstructural analysis” should be used.

Author Response

Point 1: Line 85: It would be more transparent if the title of the chapter would indicated which material the testing methods refer to.

Response 1: Many thanks to the reviewers for their valuable comments. The materials used in the test method have been supplemented, specifically modified as “The effects of incorporating different mass fractions of UFS, SF and PCE at different water-cement ratios on the properties of the grouting materials were investigated using orthogonal tests and polar analysis of variance, and the optimal mix ratios for the grouting materials were obtained through comprehensive equilibrium analyses, i.e., water-cement ratio of 0.70, 20% (mass fraction) of UFS, 12% (mass fraction) of SF, and 0.16% (mass fraction) of PCE”. Please refer to the revised draft for details. The modified part has been marked in red.

Point 2: Line 81: “%” is mass %? Please specify (Point 3 in previous MS version).

Response 2: Many thanks to the reviewers for their valuable comments. “%” has been removed and Table 1 has been modified to add “Mass fraction /%” in the corresponding position. Please refer to the revised draft for details. The modified part has been marked in red.

Point 3: Line 91: Please unify the capitalization in Table 2 and check the units for fluidity and viscosity.

Response 3: Many thanks to the reviewers for their valuable comments. It has been modified in the revised draft, see the revised draft for details, and the modified part has been marked in red. Fluidity and viscosity test is based on “Technical Specifications for Cement grouting Construction of hydraulic buildings”, the specific steps have been introduced in section 2.3.

Point 4: The chapter title 2.3.5 “Microscopic test” is an unusual expression, as well as XRD and TGA are not performed by microscope.  It would be maybe better to say “Mineralogical and microstructural analysis” or similar. Similar comment for 3.6 Microstructure analysis (line 262), where it would be also more correct to say “Mineralogical and microstructural analysis”. Line 332: At point (3) at the beginning of the sentence instead of »Microscopic analysis.« the term “Mineralogical and microstructural analysis” should be used.

Response 4: Thank you very much for the valuable comments of the reviewers. The corresponding position has been modified, as detailed in the revised draft, and the modified part has been marked in red.

Point 5:  For XRD please add main instrument setup and sample preparation (were the samples dry, granulometry of analysed samples) and  list the powder diffraction file (PDF) codes used for phase determination.  For SEM please add the instrument information (producer, model) and main settings (filament, vacuum mode (low or high vacuum) an accelerating voltage, working distance etc.) and sample preparation (any coating?).

Response 5: Thank you very much for the valuable comments of the reviewers. The corresponding equipment information and sample preparation have been added in the revised draft, see the revised draft for details, and the modified part has been marked in red.

Point 6: Point 7: Figure captions of Figures 2-5 (in the revised version) were not corrected.

Response 6: Thank you very much for the valuable comments of the reviewers. The title in Figure 2-5 has been revised, please see the revised draft for details. The modified part has been marked in red.

Point 7: Line 266: Please add some discussion (comparison with other authors) for the sentence “Following freezing and thawing in aqueous solution, Figure 7 shows that the sample’s primary hydration products are Ca(OH)2, AFt (ettringite) crystal, C-S-H gel, and CaCO3.”

Response 7: Thanks to the valuable comments of the reviewers, the corresponding literature has been added to the revised draft, and the revised part has been marked in red.

Point 8: Line 280: Figure 7 captions should be improved. In legend either chemical formula either phase name should be used (now is mixed), please unify. XRD peaks of gypsum are not clear. The diagram might be cut at 55 or 60°2 theta in order to make peaks more clear.

Response 8: Thank you very much for the valuable comments of the reviewers. The title of Figure 7 has been modified, the chemical formula and phase name in the figure have been unified, and the figure has been modified again. See the revised draft for details.

Point 9: Point 10: Based on SEM micrographs from Figure 9 it is still hard to identify phases. Some labels are incomprehensible (different alphabet) and should be corrected.

Response 9: The SEM images have been adjusted, and the title of Figure 9 has been explained in detail. For details, see the revised draft. The modified part has been marked in red.

Reviewer 2 Report

The authors incorporated most of the reviewers' comments improving the manuscript in the process however, there are still some improvements to be made prior publication, mainly the language edditing as in some cases the ideas are not easy to catch, especially for non-native speakers.

added text L188-192 - there are units missing

Figure 9 still contains chinese symbols reffering to cracks/pores in the SEM images.

Conclusions should not use abbreviations not specified in conclusions, thus use gypsum instead of Gyp.

Conflict of interest is copy of data availability statement, please revise.

Please check and rephrase the text for 'tortured sentences', comprehensibility of the statements etc. There seems to be no edditing done inbetween the revisions.

Author Response

Point 1: The authors incorporated most of the reviewers' comments improving the manuscript in the process however, there are still some improvements to be made prior publication, mainly the language edditing as in some cases the ideas are not easy to catch, especially for non-native speakers.

Response 1: Thank you very much for the valuable comments of the reviewers, and the language of the revised draft has been modified and polished.

Point 2: added text L188-192 - there are units missing. Figure 9 still contains chinese symbols reffering to cracks/pores in the SEM images.

Response 2: Thank you very much for the valuable opinions of the reviewers. The units have been added in the corresponding positions and the SEM photos in Figure 9 have been modified. For details, see the revised draft, and the modified parts have been marked in red.

Point 3: Conclusions should not use abbreviations not specified in conclusions, thus use gypsum instead of Gyp.

Response 3: Thanks very much for the valuable comments of the reviewers, the abbreviation “Gyp” in the conclusion (3) has been changed to “CaSO4•2H2O”, as detailed in the revised draft, and the modified part has been marked in red.

Point 4: Conflict of interest is copy of data availability statement, please revise.

Response 4: Thank you very much for the valuable opinions of the reviewers. The conflict of interest has been revised in the revised draft, and the modified part has been marked in red.
